# AlgoForge: Specializing Code Generation Agents through Collaborative Reinforcement Learning

## Abstract

Large language models (LLMs) have achieved impressive results in code generation across many programming tasks. However, most existing approaches rely on autoregressive decoding without global planning, often yielding locally coherent but globally suboptimal solutions, i.e., code that may fail to pass all test cases or incur unnecessary time or space complexity. Recent efforts, such as Chain-of-Thought (CoT) and multi-agent system (MAS) paradigms, introduce a planning stage, but their limited role specialization and coordination reduce effectiveness on complex tasks. In this work, we present AlgoForge, a collaborative code generation framework that integrates two specialized LLM agents, a Planner and a Coder, to jointly perform plan-to-code translation. We first construct two dedicated cold-start datasets, the Planner Dataset and the Coder Dataset, to inject algorithmic knowledge and instruction-following skills into each agent via supervised fine-tuning. Building upon this initialization, we further enhance both agents through a collaboration-aware reinforcement learning stage based on Gradient-based Reinforcement Policy Optimization (GRPO), enabling stronger specialization and alignment. We evaluate AlgoForge on four benchmarks of varying difficulty (LiveBench, MBPP, CodeContests, and CodeForces) using three base models (Qwen2.5-7B-Instruct, Qwen2.5-7B-Coder-Instruct, and Qwen2.5-14B-Coder-Instruct). AlgoForge consistently outperforms the base models, improving Pass@1 by up to 12.2% on MBPP and 36.5% on CodeContests, while also reducing time and space complexity, as well as lowering failure rates and improving runtime efficiency and maintainability. These results demonstrate the effectiveness of combining role specialization with collaborative reinforcement learning for robust LLM-based code generation.

## 1 Introduction

With the rapid development of large language models (LLMs) in recent years, LLM-powered code generation methods have shown remarkable capabilities in a wide range of code generation tasks (Liu et al., 2024; Fried et al., 2023; Koziolek et al., 2024; Li et al., 2024; AlOmar et al., 2024). LLMs with advanced reasoning abilities, including DeepSeek (Guo et al., 2025; 2024), LLaMA (Touvron et al., 2023), Qwen (Hui et al., 2024), and GPT (Achiam et al., 2023), have achieved notable results on a wide range of code generation benchmarks. During the generation process, these models typically follow an autoregressive decoding strategy, predicting the next token one step at a time based on the previously generated tokens. This sequential generation mechanism can be viewed as a token-level Markov process (Liu et al., 2025; Wan et al., 2025; Yao et al., 2023). However, such a generation mechanism may inherently lack global planning. As a result, it often produces outputs that, while locally coherent, are suboptimal solutions from a global perspective.

To improve the accuracy of code generation, prior studies have proposed the Chain-of-Thought (CoT) method (Wei et al., 2022), which introduces reasoning steps or pseudocode before generation to enhance planning and reduce errors. However, CoT falls short when dealing with complex, multi-faceted tasks. To address this, researchers have further explored multi-agent systems (MAS) based on large language models, which simulate roles such as requirements engineers, programmers, and testers to collaboratively accomplish end-to-end software development workflows (Islam

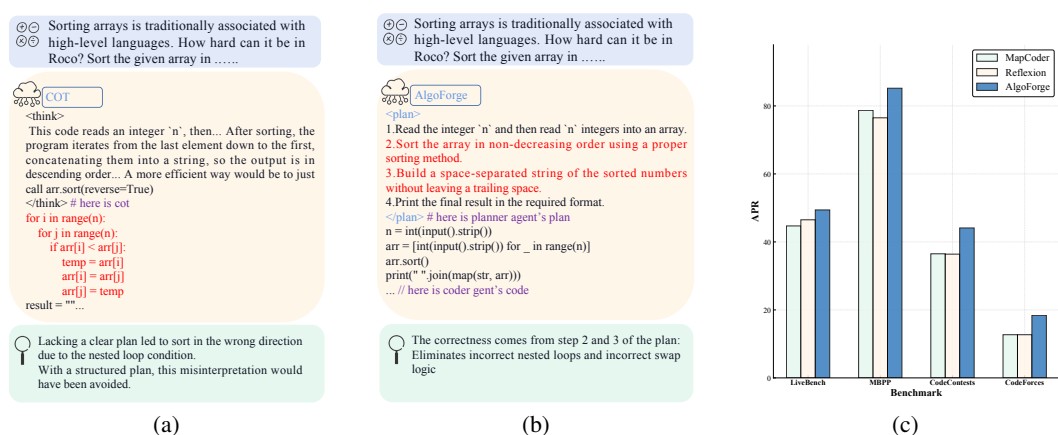

Figure 1: (a)CoT for code generation. (b) AlgoForge for code generation. (c) A comparison with different multi-agent systems for code generation, where Qwen2.5-7B-Coder-Instruct serves as the baseline model. More example can be found in Appendix 4.

et al., 2024; Lin et al., 2025). A representative approach in this line of work is the Planner–Coder paradigm (Li et al., 2025), which decomposes code generation into separate planning and coding stages, each handled by a distinct agent. This framework draws inspiration from the human practice of first devising a high-level strategy before actual programming, yet it also faces limitations: without sufficient specialization in the agents, its overall performance gains remain constrained. For example, SCOT, when evaluated on Qwen2.5-7B-Instruct, achieves only marginal improvements on the CodeContests benchmark (see Table 2), and assessments by GPT-o3 indicate that about one-third of its algorithmic thoughts are substandard. Even valid plans often fail to effectively guide the model, thereby diminishing the paradigm's effectiveness when the base model lacks strong reasoning and interpretive capabilities.

To address this issue, we propose the AlgoForge framework, which integrates a specialized Planner and Coder to collaboratively perform code generation. We first construct two dedicated datasets, namely the Planner Agent Dataset and the Coder Agent Dataset, to provide targeted supervision for each agent. Using these datasets, we perform a cold start initialization based on supervised fine tuning (SFT), enabling the Planner to generate structured algorithmic plans and the Coder to accurately translate them into executable code. Inspired by prior work (Li et al., 2025; Le et al., 2024; Chen et al., 2023b), we decompose algorithmic reasoning into four components, which includes input–output definition, linear progression, conditional logic, and iteration—providing explicit guidance for both agents. After cold start initialization, we further enhance their specialization through a collaboration-aware reinforcement learning stage. Building on the strong performance of the Gradient-based Reinforcement Policy Optimization (GRPO) algorithm (Shao et al., 2024) in LLM reasoning, we design a collaborative GRPO framework in which both agents are optimized jointly. This joint optimization strengthens their individual specialization while improving communication and alignment, enabling them to work together toward generating efficient code.

We evaluate AlgoForge on LiveBench, MBPP, CodeContests, and CodeForces using Qwen2.5-7B-Instruct, Qwen2.5-7B-Coder-Instruct, and Qwen2.5-14B-Coder-Instruct. Across Pass@1/5 and average pass rate, AlgoForge yields large gains—e.g., vs. Qwen2.5-7B-Instruct: +12.2% (MBPP) and +36.5% (CodeContests); vs. Qwen2.5-7B-Coder-Instruct: +9.5% (MBPP) and +31.5% (Code-Contests). It also lowers runtime, memory use, cyclomatic complexity, and failure rates, improving efficiency and maintainability.

Our contributions are as follows:

- **Collaborative Code Generation Framework**: We propose AlgoForge, a novel framework with a specialized Planner and Coder, where each agent is first initialized via supervised fine- tuning with algorithmic reasoning and then enhanced through reinforcement learning, enabling more effective plan-to-code translation.

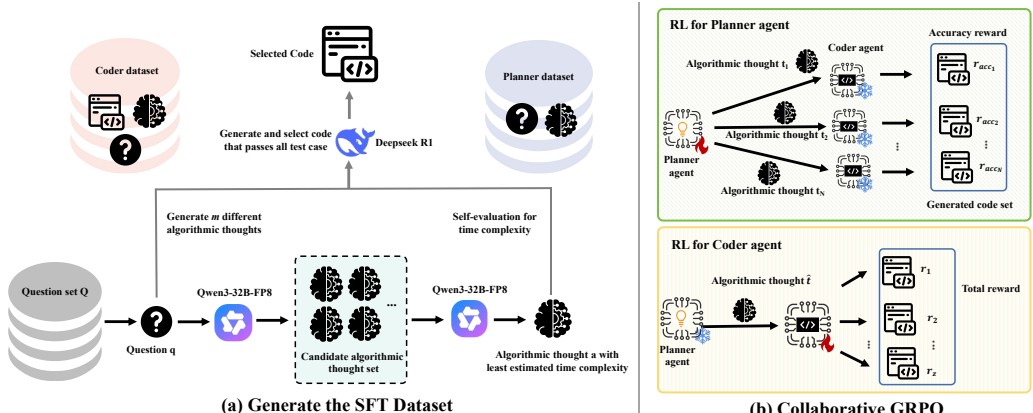

Figure 2: Overview of AlgoForge.

- **Construct code generation knowledge datasets**: For the cold-start phase, we construct two dedicated datasets, the Planner Dataset and the Coder Dataset, which provide knowledge of cold-start initialization for both agent.

- **Collaboration-Aware Reinforcement Learning**: We design a collaborative GRPO-based reinforcement learning method to jointly optimize both agents, enhancing their specialization and coordination for accurate and efficient code generation.

- **Comprehensive experiments and evaluations**: We conducted extensive experiments with AlgoForge on four different datasets: LiveBench (White et al., 2024), MBPP (Austin et al., 2021), CodeContests (Li et al., 2022), and CodeForces (Quan et al., 2025). The results demonstrate that AlgoForge not only achieves state-of-the-art performance but also significantly enhances the runtime efficiency and maintainability of the generated code.

## 2 RELATED WORK

Due to space constraints, additional related work is provided in Appendix B.1.

### 2.1 LLM BASED CODE GENERATION

The automatic generation of program code or completion of snippets from natural language using LLMs has gained much attention (Guo et al., 2024; Wei et al., 2022; Zhang et al., 2023; Islam et al., 2024; Jiang et al., 2024; Izadi et al., 2024; Lin et al., 2025; Zhang et al., 2025c;a), improving efficiency and reducing human error (Huang et al., 2023b; Geng et al., 2024). Models such as GPT-4o (Achiam et al., 2023), ChatGLM (GLM et al., 2024), CODEX (Pasquini et al., 2010), Qwen (Hui et al., 2024), DeepSeek (Guo et al., 2025), and CodeGen (Nijkamp et al., 2022) show strong code generation and understanding abilities, achieving SOTA results on MBPP (Austin et al., 2021) and HumanEval (Chen et al., 2021). Their success stems from large-scale training (Lozhkov et al., 2024) and SFT for better coding abilities (Chang et al., 2024).

Prompt-based methods further enhance performance. CoT (Wei et al., 2022) generates reasoning steps to guide code; retrieval-based prompting incorporates relevant examples (Nashid et al., 2023; Kang et al., 2023); ChatUniTest locates focal methods for test generation (Xie et al., 2023); prompt composition adds high-level descriptions before code generation (Yuan et al., 2024); and CodeT leverages self-generated tests (Chen et al., 2023a).

## 3 APPROACH

In this section, we present the methodology of the AlgoForge framework, which consists of two stages: *Customized Cold-Start Initialization* and a subsequent post-training process. The Cold-Start Initialization is based on supervised fine-tuning (SFT), where we construct two datasets to enhance

each agent's specialization: the **planner dataset** for algorithmic planning and the **coder dataset** for code generation based on the plans.

Building on this, we introduce a collaboration-aware reinforcement learning method called **collaborative GRPO**. This two-stage GRPO algorithm jointly optimizes the planner and coder, encouraging them to collaborate toward a unified objective and improving both accuracy and efficiency in code generation. Specifically, the planner is optimized via RL to produce more efficient algorithmic plans, while the coder is rewarded for faithfully translating these plans into correct code, ensuring alignment between the two agents. The overall process is illustrated in Fig.2.

### 3.1 CUSTOMIZED COLD-START INITIALIZATION FOR PLANNER AND CODER SPECIALIZATION

#### 3.1.1 SFT DATASET CONSTRUCTION

To inject foundational knowledge and enhance the specialization of both the planner and the coder, we construct two distinct SFT datasets, denoted as $S_{\text{planner}}$ and $S_{\text{coder}}$, each tailored to the unique roles and capabilities of the planner and the coder. To support this, we curated a collection of 15,000 programming questions from various datasets (Xu et al., 2025; Guha et al., 2025; Li et al., 2023), which serve as the foundation for building the AlgoForge's SFT datasets.

For each question $q$, the dataset provides not only the programming prompt but also an accompanying set of test cases, which serve to evaluate and validate the correctness of the generated solution.

**Construction of the SFT Dataset for Planner Specialization:** The Planner dataset is a curated collection of high-quality algorithmic thought examples, distilled to inject structured and effective algorithmic thinking into the planner agent during SFT process. The dataset consists of $n$ pairs of questions and corresponding algorithmic thoughts, denoted as $S_{\text{planner}} = \{(q_i, t_i)\}_{i=1}^n$. In this context, $q_i$ represents a chosen algorithmic question, while $t_i$ refers to the distilled algorithmic thought for each $q_i$, generated by more advanced LLMs. In this paper, given the impressive performance of Qwen3-32B-FP8 (Yang et al., 2025a) on code generation benchmarks and its strong logical reasoning ability, we select it as the distillation model to generate algorithmic thoughts $t_i$ for each corresponding question $q_i$.

Inspired by (Li et al., 2025; Le et al., 2024; Chen et al., 2023b), our algorithmic thought is organized into four core components: input-output definition, linear progression, conditional logic, and iteration. The input-output definition clarifies the function signature by specifying the given information and the expected outcome. Linear progression captures the fundamental step-by-step operations that transform inputs into outputs. Conditional logic governs decision-making by selecting different execution paths under varying conditions. Iteration models repeated computation over data structures or ranges. This structured formulation aligns closely with programming semantics, promotes interpretability, and facilitates the generation of accurate and coherent code. Fig 3 shows an example of algorithmic thoughts along with the prompt produced by the Planner Agent.

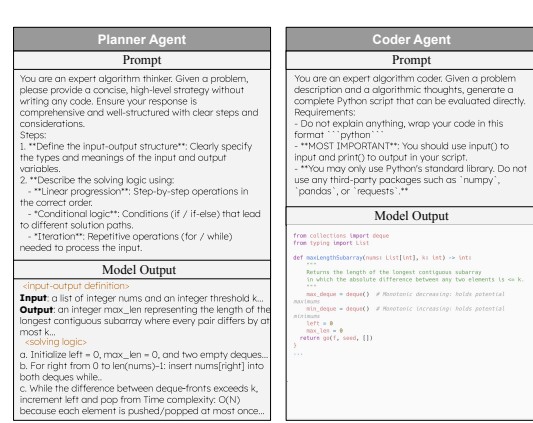

Figure 3: Algorithmic thoughts: Planner Agent prompt is on the left and the Coder Agent prompt is on the right.

To ensure the generation of computationally efficient code with minimal time complexity, we incorporate a self-evaluation mechanism that assesses algorithmic thoughts based on their estimated complexity. For example, given a single question, multiple algorithmic thoughts are generated. Each thought is then evaluated for its complexity (e.g., in Big-O notation) by the same LLM, and the one with the lowest predicted complexity is selected. Specifically, for a given question $q$, we generate multiple candidate algorithmic thoughts $\{\hat{t}_1, \hat{t}_2, \ldots, \hat{t}_m\}$, each representing a different approach to

solving $q$. We then perform self-evaluation on each candidate using the same LLM (e.g., Qwen3-32B-FP8), automatically estimating its time complexity $\mathcal{T}(\hat{t}_j)$. Among all candidates, we select the one with the lowest estimated complexity, i.e., $t = \arg\min_{\hat{t}_j} \mathcal{T}(\hat{t}_j)$, and pair it with the original question $q$ to form a data point $(q, t)$.

For the planner dataset, it consists of algorithmic thoughts $t$ distilled from a powerful model, which serve to inject algorithmic knowledge into the planner agent in order to provide better guidance for the coder agent.

**Construction of the SFT Dataset for Coder Specialization:**

The Coder dataset consists of triplets, denoted as $S_{\text{Coder}} = \{(q_i, t_i, c_i)\}_{i=1}^n$, where $q_i$ is a selected algorithmic question and $t_i$ is the corresponding algorithmic thought. $c_i$ represents the ground-truth code derived from the algorithmic thought $t_i$ and the corresponding question $q_i$. Specifically, for each given question $q$ together with its associated algorithmic thoughts $t$, we use the DeepSeek R1 API (Guo et al., 2025) to generate multiple code snippets $c_i$, and then select those that successfully pass all test cases provided with the question. Each $c_i$ faithfully follows the reasoning encoded in $t_i$ and successfully satisfies all accompanying test cases, thereby qualifying as a correct solution. By aligning structured algorithmic reasoning with executable and verifiable code, the *Coder dataset* bridges high-level planning and practical implementation. This alignment enables the coder agent to learn how to faithfully follow high-level algorithmic thoughts and translate them into accurate, functional code implementations.

### 3.1.2 SFT Optimization of Planner and Coder Agents

After constructing both the planner dataset and the coder dataset, we initialize both the planner agent and the coder agent based on the initial LLM $\pi_{\theta_{\text{init}}}$ through a SFT process, where $\theta_{\text{init}}$ denotes the parameters of the pretrained initial LLMs. This initialization serves as a foundational adaptation process, enabling each agent to specialize in its respective role. In the following, we will provide a detailed explanation of the optimization process of both agents.

**SFT Optimization for Planner Agents:** In this stage, starting from the initial LLMs $\pi_{\theta_{\text{init}}}$, we expect the planner agent to acquire specialized and foundational knowledge and capabilities, so that it can design algorithmic thoughts $a$ specifically tailored to given problems $q$. The optimization objective for the planner agent is to find the parameter $\theta_{\text{planner}}^*$ with planner dataset $S_{\text{planner}}$:

$$\theta_{\text{planner}}^* = \arg\min_{\theta_{\text{planner}}} -\mathbb{E}_{(q_i, t_i) \sim S_{\text{planner}}} \left[ \sum_{i=1}^n \log \pi_{\theta_{\text{planner}}}(t_i \mid q_i) \right], \tag{1}$$

where $\theta_{\text{planner}}^*$ represents the optimal parameters of the planner agent obtained through cold-start optimization, starting from pretrained initialization $\theta_{\text{init}}$. After cold-start optimization, $\pi_{\theta_{\text{planner}}}$ acquires an initial capability to generate algorithmic thoughts that are well-aligned with problem instances.

**SFT Optimization for Coder Agents:** The coder agent $\pi_{\theta_{\text{coder}}}$ learns to generate the correct and executable code $c$ for a given problem $q$, guided by algorithmic thoughts $a$ as an intermediate bridge. Here, $\theta_{\text{coder}}$ denotes the trainable parameters that govern the agent's code generation behavior. Specifically, it learns the mapping $\pi_{\theta_{\text{coder}}}(q, a) \to c$. Therefore, the optimization objective of $\theta_{\text{coder}}$ can be formulated as:

$$\theta_{\text{coder}}^* = \arg\min_{\theta_{\text{coder}}} -\mathbb{E}_{(q_i, t_i, c_i) \sim S_{\text{coder}}} \left[ \sum_{i=1}^n \log \pi_{\theta_{\text{coder}}}(c_i \mid q_i, t_i) \right]. \tag{2}$$

$\theta_{\text{coder}}^*$ represents the optimal parameters that enable the coder agent to generate correct code given a problem and the corresponding algorithmic thoughts. This optimization empowers the coder agent to comprehend algorithmic thoughts and effectively generate accurate and executable code accordingly.

### 3.2 Collaborative GRPO Reinforcement Learning Framework

After the cold-start initialization phase, in which the planner and coder acquire foundational knowledge and role-specific expertise, we introduce **Collaborative GRPO**, an extension of the GRPO

reinforcement learning algorithm (Guo et al., 2025), to jointly optimize both agents and further strengthen their specialized capabilities. In this paper, we adopt GRPO as the backbone of our approach in the code generation domain, owing to its strong reasoning capabilities and its effectiveness in tackling complex, multi-step decision-making problems (Shao et al., 2024; Guo et al., 2025). Our collaborative GRPO framework consists of two RL stages that specialize the Planner and Coder agents. First, the Planner Specialization stage uses collaboration-aware reinforcement learning to train the Planner to generate structured, executable algorithmic thoughts. Then, the Coder Specialization stage reinforces the Coder to accurately interpret and implement these thoughts. This two-stage process enables role decoupling and collaboration, improving both code accuracy and efficiency.

### 3.2.1 RL FOR PLANNER AGENT SPECIALIZATION

Planner agent specialization falls under the collaborative RL paradigm. As the name suggests, its training process is based on the cooperative interaction between the planner and coder agents, who work together to optimize the parameters of the planner agent $\theta_{\text{coder}}$.

Given a question $q$, planner agent first generate $N$ distinct algorithmic thoughts $\{t_i\}_{i=1}^N$, where $t_i = \pi_{\theta_{\text{planner}}}(q)$. To evaluate the quality of each algorithmic thought, we adopt an indirect metric: the accuracy of code generated by the Coder Agent under the policy $\pi_{\theta_{\text{coder}}}$. Specifically, for a given thought $t_i$, we pair it with the question $q$ and input them into the Coder Agent to produce $M$ candidate code snippets $\{c_{i,j}\}_{j=1}^M$, where each code snippets is generated as $c_{i,j} = \pi_{\theta_{\text{coder}}}(t_i, q)$. The more of the question's provided test cases that the set $\{c_{i,j}\}_{j=1}^M$ successfully passes, the higher the quality we ascribe to the corresponding algorithmic thought $t_i$. Therefore, the accuracy reward $r_{\text{acc}_i}$ for algorithmic thought $t_i$ can be expressed as:

$$r_{\text{acc}_i} = \frac{1}{M} \sum_{j=1}^{M} \text{softmax}(p_{i,j}), \tag{3}$$

where $p_{i,j}$ denotes the proportion of test cases passed by the code snippet $c_{i,j}$, the accuracy reward $r_{\text{acc},i}$ aggregates the performance of multiple candidate code snippets derived from the same algorithmic thought $t_i$. Rather than simply averaging the raw pass rates $p_{i,j}$, we apply a softmax-based weighting to emphasize higher-quality snippets that succeed on more test cases. Algorithmic thoughts that lead to superior code receive higher rewards, while those producing weaker solutions are down-weighted. This design provides a more informative and discriminative reward signal for evaluating the quality of algorithmic reasoning. After obtaining the set of accuracy rewards $\{r_{\text{acc},i}\}_{i=1}^n$ for all candidate thoughts, we compute the advantage function using Eq. 7 and update the planner parameters $\pi_{\theta_{\text{planner}}}$ according to Eq. 6.

During this RL stage, the Planner and Coder Agents collaborate to generate reward signals, used solely to update the Planner's parameters. The Planner is optimized to yield higher-quality algorithmic thoughts, enabling the Coder to produce more accurate code. The Coder's parameters remain frozen at this stage. In the subsequent RL stage for Coder specialization, its parameters are updated, as detailed in Section 3.2.2.

### 3.2.2 RL FOR CODER AGENT SPECIALIZATION

After RL for planner specialization, the specialized planner agent becomes capable of generating higher-quality algorithmic thoughts. In the subsequent stage, the focus of RL shifts to coder agent specialization, aiming to enhance the coder agent's ability to effectively follow these algorithmic thoughts and produce accurate, efficient code.

For a given question $q$, we first use the reinforcement-trained planner agent to generate an algorithmic thought $\hat{t}$, where $\hat{t} = \pi_{\theta_{\text{planner}}}(q)$. Guided by the algorithmic thought $\hat{t}$, the coder agent generates a set of code snippets $c_{i_{i=1}}^z$, consisting of $z$ code snippets. This process can be formulated as $\{c_i\}_{i=1}^z = \pi_{\theta_{\text{coder}}}(q, \hat{t})$. In a similar manner, based on the test case pass rate defined in Eq. 3, we calculate the accuracy reward for each code snippet, resulting in the reward set $\{r_{\text{acc}_i}\}_{i=1}^z$.

The objective of the coder agent is not only to generate accurate code, but also to produce efficient implementation. For a set of code snippets $\{c_i\}_{i=1}^z$, we assign a memory efficiency reward if at least one code snippet passes all test cases; if no snippet passes, the efficiency reward is set to $0$. We

employ package `psutil` to monitor both storage space and memory usage. Among all snippets that pass every test case, the one with the smallest memory consumption is defined to have the target memory usage, denoted as $\mathcal{O}_{\text{target}}$. The value of each space efficiency reward $r_{\text{space}_i}$ of code snippet $c_i$ is determined as:

$$r_{\text{space}_i} = \begin{cases} \exp\left(-\left|\mathcal{O}(c_i) - \mathcal{O}_{\text{target}}\right|\right), & \text{if } c_i \text{ passes all test cases,} \\ 0, & \text{otherwise.} \end{cases} \tag{4}$$

where $\mathcal{O}(c_i)$ represents the space complexity of the code snippet $c_i$. The space efficiency reward is only applicable when the generated code snippet passes all test cases. Its purpose is to ensure that while rewarding the accuracy of code generation, the coder agent is also gradually encouraged to focus on producing implementations whose space complexity approaches $\mathcal{O}_{\text{target}}$. Finally, our total reward $r_i$ for a code snippet is defined as:

$$r_i = r_{\text{acc}_i} + \lambda r_{\text{space}_i}, \tag{5}$$

where $\lambda$ represents the hyperparameter. After obtaining the total reward $r_i$, we compute the advantage function based on Eq. 7, and update the parameters of the coder agent according to Eq. 6.

Table 1: Performance on LiveBench and MBPP using Pass@1 (↑), Pass@5 (↑), and APR (↑). **Bold** is best per block.

| Base Model | Method | LiveBench | | | MBPP | | |
|---|---|---|---|---|---|---|---|
| | | Pass@1 | Pass@5 | APR | Pass@1 | Pass@5 | APR |
| Qwen2.5-7B-Instruct | - | 28.3 | 35.2 | 41.9 | 63.6 | 68.7 | 71.4 |
| Qwen2.5-7B-Coder-Instruct | - | 32.8 | 37.5 | 44.5 | 67.7 | 74.4 | 77.2 |
| Qwen2.5-14B-Coder-Instruct | - | 40.6 | 47.7 | 53.1 | 78.3 | 81.9 | 85.0 |
| DeepSeek-Coder-V2-16B | - | 23.4 | 26.4 | 36.5 | 71.9 | 77.8 | 77.9 |
| Qwen2.5-7B-Instruct | CoT | 29.3 | 36.6 | 44.5 | 64.2 | 67.2 | 73.9 |
| | SCoT | 31.3 | 36.5 | 43.9 | 65.7 | 69.5 | 75.1 |
| | MapCoder | 33.9 | 36.5 | 43.2 | 64.7 | 69.5 | 70.7 |
| | Reflexion | 33.6 | 35.9 | 44.2 | 64.3 | 70.4 | 72.2 |
| | GRPO | 33.2 | 37.7 | 42.5 | 69.4 | 73.7 | **80.9** |
| | AlgoForge | **37.4** | **44.9** | **47.5** | **71.4** | **74.9** | 77.2 |
| Qwen2.5-7B-Coder-Instruct | CoT | 33.2 | 39.1 | 42.9 | 68.2 | 74.8 | 77.4 |
| | SCoT | 36.3 | 42.2 | 44.3 | 68.4 | 74.3 | 78.2 |
| | MapCoder | 34.7 | 43.1 | 44.9 | 68.9 | 76.9 | 78.7 |
| | Reflexion | 32.7 | 39.9 | 46.5 | 69.7 | 74.9 | 76.5 |
| | GRPO | 34.5 | 41.2 | **51.3** | 73.4 | 76.7 | 82.5 |
| | Reasonflux-Coder | 37.1 | 39.3 | 48.7 | 70.2 | 74.5 | 78.7 |
| | AlgoForge | **40.2** | **45.5** | 49.4 | **74.3** | **78.5** | **85.2** |
| Qwen2.5-14B-Coder-Instruct | CoT | 41.4 | 49.3 | 53.8 | 79.5 | 83.5 | 85.7 |
| | SCoT | 39.6 | 41.1 | 42.9 | 77.4 | 79.7 | 83.8 |
| | MapCoder | 41.9 | 47.5 | 50.7 | 78.9 | 82.0 | 84.7 |
| | Reflexion | 42.3 | 48.6 | 50.5 | 80.5 | 82.4 | 85.5 |
| | GRPO | 45.5 | 50.7 | 55.4 | 80.7 | 83.2 | 85.2 |
| | Reasonflux-Coder | 47.5 | 49.3 | **56.7** | 78.5 | 80.5 | 81.9 |
| | AlgoForge | **48.7** | **51.3** | 56.1 | **82.1** | **84.4** | **86.5** |

# 4 EXPERIMENT

## 4.1 EXPERIMENT SETUP

### 4.1.1 BASE MODEL AND BENCHMARK

Since AlgoForge requires additional fine-tuning and training of LLMs, this paper focuses on open-source models. In our experiments, we choose Qwen2.5-7B-Instruct (Yang et al., 2025b), Qwen2.5-7B-Coder-Instruct and Qwen2.5-7B-Coder-Instruct (Hui et al., 2024) as the base models for AlgoForge. We evaluate them on four different benchmarks: LiveCode (White et al., 2024), MBPP (Austin et al., 2021), CodeContests (Li et al., 2022) and CodeForces (Quan et al., 2025). LiveCode and MBPP are classified as basic function-level programming tasks, which focus on evaluating the fundamental programming skills of a model, while CodeContests and CodeForces are classified as complex programming tasks of competition level, which focus on evaluating the advanced algorithm design and complexity management capabilities of a model.

### 4.1.2 METRICS

For code generation accuracy, we adopt Pass@1, Pass@5, and Average Pass Rate (APR) (Li et al., 2025; Wang et al., 2025a; Zhang et al., 2025d). Pass@1 checks correctness on the first attempt, Pass@5 allows up to five attempts, and APR measures the average proportion of passed test cases. For efficiency, we record average runtime and memory usage (MU, in char). For maintainability and correctness, we compute average cyclomatic complexity (CC) and failure rate (FR). Finally, we measure inference time during code generation.

Table 2: Performance on CodeContests and CodeForces using Pass@1 ($\uparrow$), Pass@5 ($\uparrow$), and APR ($\uparrow$). **Bold** is best per block.

| Base Model | Method | CodeContests | | | CodeForces | | |
|---|---|---|---|---|---|---|---|
| | | Pass@1 | Pass@5 | APR | Pass@1 | Pass@5 | APR |
| Qwen2.5-7B-Instruct | - | 19.4 | 21.9 | 27.2 | 5.6 | 5.9 | 7.7 |
| Qwen2.5-7B-Coder-Instruct | - | 21.3 | 24.4 | 33.9 | 6.2 | 8.3 | 13.4 |
| Qwen2.5-14B-Coder-Instruct | - | 29.1 | 33.1 | 41.4 | 9.2 | 13.5 | 17.5 |
| DeepSeek-Coder-V2-16B | - | 21.3 | 28.5 | 34.4 | 7.5 | 9.6 | 16.7 |
| Qwen2.5-7B-Instruct | CoT | 20.9 | 22.7 | 30.5 | 5.6 | 6.0 | 7.9 |
| | SCoT | 22.9 | 25.5 | 33.2 | 6.7 | 7.2 | 12.6 |
| | MapCoder | 21.7 | 27.3 | 42.5 | 7.2 | 7.7 | 13.3 |
| | Reflexion | 22.7 | 25.5 | 33.6 | 7.5 | 7.9 | 11.4 |
| | GRPO | 20.7 | 26.4 | 29.2 | 6.4 | 6.9 | 12.6 |
| | AlgoForge | **26.5** | **33.2** | **39.4** | **8.5** | **10.4** | **14.4** |
| Qwen2.5-7B-Coder-Instruct | CoT | 23.7 | 26.0 | 34.1 | 6.4 | 8.7 | 13.8 |
| | SCoT | 24.2 | 27.9 | 36.5 | 7.7 | 9.6 | 14.4 |
| | MapCoder | 24.0 | 26.1 | 36.5 | 7.3 | 6.9 | 10.2 |
| | Reflexion | 22.8 | 29.5 | 36.4 | 8.0 | 8.2 | 12.7 |
| | GRPO | 27.2 | 31.5 | **46.3** | 9.1 | 9.7 | 13.3 |
| | Reasonflux-Coder | 25.9 | 31.7 | 39.2 | 8.2 | 9.4 | 11.9 |
| | AlgoForge | **29.3** | **33.2** | 44.1 | **12.3** | **15.9** | **18.4** |
| Qwen2.5-14B-Coder-Instruct | CoT | 28.7 | 29.9 | 37.4 | 10.4 | 11.8 | 17.8 |
| | SCoT | 31.3 | 33.8 | 43.2 | 10.1 | 11.4 | 17.2 |
| | MapCoder | 30.9 | 32.1 | 39.7 | 9.7 | 10.4 | 14.9 |
| | Reflexion | 32.2 | 34.7 | 40.4 | 11.3 | 12.9 | 15.5 |
| | GRPO | 33.7 | 35.9 | 42.3 | 12.9 | 14.3 | 17.6 |
| | Reasonflux-Coder | 32.1 | 37.2 | 43.6 | 12.1 | 14.1 | 18.8 |
| | AlgoForge | **35.9** | **41.2** | **44.5** | **13.5** | **16.7** | **18.7** |

### 4.1.3 BASELINES

Under the same base model configuration, we selected four prompt- based methods (CoT (Wei et al., 2022), SCoT (Li et al., 2025), MapCoder (Islam et al., 2024), and Reflexion (Shinn et al., 2023)) and two RL-based methods (GRPO (Shao et al., 2024) and Reasonflux-Coder (Wang et al., 2025a)) as baselines for comparison. GRPO is a novel RL algorithm designed to enhance the reasoning ability of LLMs. In this paper, we adopt the reward function proposed in (Robeyns & Aitchison, 2025) to optimize the base model via RL. To ensure a fair and consistent comparison, we use the same RL dataset as AlgoForge.

### 4.2 EXPERIMENTAL RESULTS

Due to space limitations, we placed the evaluation of AlgoForge regarding efficiency and maintainability, error rate, and sensitive analysis in Appendix B.3.2.

**Main results:** Table 1 and Table 2 present results on LiveBench, MBPP, CodeContests, and CodeForces. Our method AlgoForge consistently outperforms both prompting-based strategies (CoT, SCoT, Reflexion, MapCoder) and advanced approaches (GRPO, Reasonflux-Coder), achieving the best or near-best performance in Pass@1, Pass@5, and APR across all base models. Notably, on Qwen2.5-7B-Instruct, AlgoForge improves LiveBench Pass@1 by +9.1 points and raises MBPP to coder-level performance, while on Qwen2.5-7B-Coder-Instruct it boosts Pass@1 by +8.0 on Code-Contests and +6.1 on CodeForces. These results demonstrate the effectiveness, robustness, and generalization ability of AlgoForge in enhancing code generation across diverse and challenging benchmarks.

**Ablation analysis:** Table 3 presents the ablation analysis of different components in AlgoForge based on Qwen2.5-7B-Coder-Instruct. Here, pt and ct respectively stand for planner training pro-

Table 3: Ablation analysis for each component of AlgoForge, where higher Pass@1 (↑), Pass@5 (↑), and APR (↑) indicate better performance. The base model considered is Qwen2.5-7B-Coder-Instruct. **Bold** indicates the best performance for clarity.

| Method | LiveBench | | | MBPP | | | CodeContests | | | CodeForces | | |
|---|---|---|---|---|---|---|---|---|---|---|---|---|
| | Pass@1 | Pass@5 | APR | Pass@1 | Pass@5 | APR | Pass@1 | Pass@5 | APR | Pass@1 | Pass@5 | APR |
| AlgoForge w/o all | 35.4 | 41.7 | 47.9 | 68.5 | 72.8 | 77.8 | 22.6 | 29.7 | 38.9 | 8.9 | 9.6 | 13.3 |
| AlgoForge (w/all) | **40.2** | **45.5** | **49.4** | **74.3** | **78.5** | **85.2** | **29.3** | **36.2** | **44.1** | **12.3** | **15.9** | **18.4** |
| **Ablations on SFT** AlgoForge w/o SFT pt | 39.4 | 44.9 | 50.7 | 72.8 | 76.2 | 83.9 | 27.9 | 34.8 | 42.7 | 11.3 | 13.9 | 16.2 |
| AlgoForge w/o SFT ct | 37.5 | 42.2 | 49.6 | 69.7 | 74.5 | 79.4 | 23.7 | 32.6 | 44.1 | 10.4 | 11.7 | 14.2 |
| AlgoForge w/o all SFT | 36.9 | 43.3 | 47.2 | 69.1 | 73.9 | 79.2 | 23.5 | 30.7 | 41.5 | 9.2 | 10.4 | 14.6 |
| **Ablations on RL** AlgoForge w/o RL pt | 38.5 | 45.2 | 48.6 | 73.3 | 77.6 | 83.2 | 28.7 | 35.4 | 43.1 | 11.4 | 13.9 | 16.5 |
| AlgoForge w/o RL ct | 37.2 | 44.9 | 47.2 | 72.2 | 74.9 | 80.5 | 26.4 | 34.3 | 41.8 | 10.3 | 12.9 | 14.9 |
| AlgoForge w/o all RL | 38.3 | 44.5 | 48.1 | 72.9 | 76.2 | 82.4 | 25.2 | 33.9 | 40.7 | 9.9 | 11.4 | 14.9 |

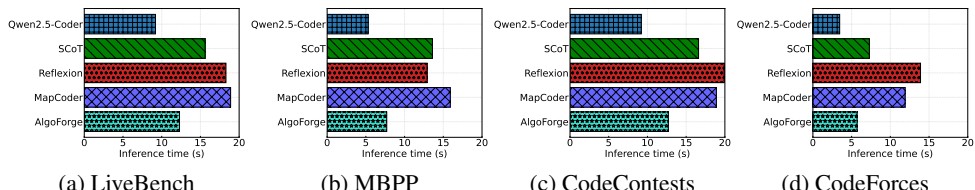

(a) LiveBench  (b) MBPP  (c) CodeContests  (d) CodeForces

Figure 4: Computation costs of different methods, where Qwen2.5-7B-Coder-Instruct is considered as base model.

cess and coder training process. Removing all components (AlgoForge w/o all) leads to a clear performance drop across all benchmarks, confirming the necessity of our design. When ablating SFT, both pre-training (AlgoForge w/o SFT pt) and continued training (AlgoForge w/o SFT ct) result in degraded performance, with the latter showing a stronger decline, especially on CodeContests and CodeForces. Similarly, ablating RL components also reduces performance, where continued training (AlgoForge w/o RL ct) is particularly critical. Overall, combining both SFT and RL yields the best performance, demonstrating their complementary contributions to AlgoForge's effectiveness.

**Inference time comparison with other multi-agent methods :** Since AlgoForge employs a collaborative planner–coder architecture, its code generation involves an extra coordination step, making runtime an important metric. We compare AlgoForge with the base model and multi-agent baselines (SCOT, Reflexion, MapCoder) using Qwen2.5-7B-Coder-Instruct. As shown in Fig. 4, Qwen2.5-7B-Coder-Instruct achieves the fastest inference (<10s), while MapCoder and Reflexion are much slower (>15s). SCOT shows intermediate latency (12–14s). AlgoForge remains efficient (8–9s), second only to Qwen2.5-7B-Coder-Instruct, highlighting its suitability for real-time use.

**RL Training Dynamics Analysis:** During the reinforcement learning process, we present the training dynamics in Figure 7, illustrating the changes in accuracy, reward, and entropy, and compare our approach against GRPO. The results demonstrate that our method consistently achieves higher accuracy and reward, while maintaining lower entropy. This indicates that under the global guidance of the planner, the coder agent is able to acquire more valuable strategies more efficiently, exhibiting greater stability and confidence in decision-making with reduced reliance on high-randomness exploration. Therefore, the planner–coder framework not only enhances task performance but also encourages the agent to develop more deterministic policy choices, leading to simultaneous improvements in both efficiency and effectiveness.

## 5 CONCLUSION

We propose AlgoForge, a collaborative framework that combines cold-start specialization and reinforcement learning to enable a Planner and a Coder agent to work together for plan-to-code translation. Experiments across multiple benchmarks show that AlgoForge consistently improves accuracy, efficiency, and maintainability over base models and existing methods, demonstrating the effectiveness of role-specialized collaboration for LLM-based code generation.

## 6 ETHICS STATEMENT

This research has been conducted in full accordance with the ICLR Code of Ethics. Throughout the study, we have maintained a responsible approach to advancing machine learning, striving not only to expand scientific knowledge but also to remain mindful of the potential societal implications of our work. Methodologically, we guarantee scientific rigor, transparency, and verifiability, and affirm that no data have been fabricated, altered, or misrepresented. Possible risks were carefully assessed during the design of the study to avoid harm to individuals or communities. In handling data, we adhered to principles of privacy protection, fairness, and inclusiveness, and ensured that all data were used under proper ethical approvals or legal licenses, with safeguards against re-identification or misuse. We acknowledge the intellectual contributions of the broader research community and provide appropriate academic credit where relevant. Overall, this work has been carried out in an open, responsible, and conscientious manner, with the goal of contributing positively to scientific progress and societal well-being, in alignment with the principles of the ICLR Code of Ethics.

## 7 REPRODUCIBILITY STATEMENT

We have taken deliberate measures to make our work reproducible. A thorough account of the experimental design—covering model structures, training strategies, and evaluation protocols—is provided both in the main text and the appendix. For the review phase, we supply an anonymous code repository link (see Appendix) that contains most of the implementation details. Upon acceptance, we will release the full source code used in all key experiments, accompanied by comprehensive documentation and step-by-step instructions so that others can reliably replicate our reported findings.

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

## A  THE USE OF LARGE LANGUAGE MODELS

A Large Language Model (LLM) was employed only for limited editorial support, such as refining wording and improving clarity of expression. The model was not involved in generating ideas, performing analyses, or contributing to the core writing of this work. All research content, interpretations, and conclusions remain solely the responsibility of the authors.

## B  APPENDIX

### B.1  RELATED WORK AND BACKGROUND

#### B.1.1  ENHANCING CODE GENERATION WITH RL AND MULTI-AGENT SYSTEMS

**RL-based enhancement:**  LLMs acquire foundational programming knowledge during pre-training, and their ability to follow instructions is further enhanced through SFT (Zhang et al., 2025c). However, prior studies have shown that the generalization ability of SFT is often limited, with models tending to overfit to training distributions and struggling on out-of-distribution tasks (Korbak et al., 2023). To further adapt these models to real-world deployment scenarios, RL serves as an effective technique, enabling them to excel in diverse and complex applications (Kumar et al., 2025).

Reinforcement Learning from Human Feedback (RLHF) (Ouyang et al., 2022), originally designed for natural language generation, has seen growing application in code generation (Zhang et al., 2024; Wang et al., 2024), where it is used to guide models toward producing outputs that better adhere to developer intentions, coding conventions, and correctness requirements. However, RLHF requires training a reward model and using Proximal Policy Optimization (PPO), a powerful RL method, which can be unstable and resource-intensive. To mitigate this, Direct Preference Optimization (DPO) (Rafailov et al., 2023) offers a simpler, more stable alternative that directly learns from human preferences without explicit reward modeling or RL. The DPO and its variants have also (Chen et al., 2024) demonstrated promising results in code generation. Chen et al. (Chen et al., 2024; Zhang et al., 2025b) introduced InfoNCA, an alignment framework that unifies the processing of explicit reward and preference data, extending the coding capabilities of DPO. Zhang et al (Zhang et al., 2025b) propose Focused-DPO, which introduces fine-grained identification and optimization of error-prone points in code. By restructuring the reward function, it emphasizes these critical segments with increased weight during training.

Recently, DeepSeek R1 (Guo et al., 2025) has demonstrated impressive reasoning capabilities on complex tasks, particularly excelling in code generation. Its RL algorithm, Gradient-based Reinforcement Policy Optimization (GRPO) (Shao et al., 2024), exhibits strong generalization and significant performance gains in complex code reasoning and generation tasks. In section B.2, we provide a detailed introduction to it.

**Multi-Agent Systems-based enhancement:**  Compared to a multi-agent system, a single LLM that generates code directly or uses pseudo-code approaches like CoT often struggles to produce complete solutions for complex problems (Islam et al., 2024). Multi-agent systems enable more flexible, efficient, and interpretable task-solving through role specialization, collaborative reasoning, and tool integration (Huang et al., 2023a). Huang et al. (Huang et al., 2023b) proposed a test executor agent that leverages a Python interpreter to generate test logs for LLMs. Similarly, Zhong et al. (Zhong et al., 2024) introduced a debugger agent that employs a static analysis tool to construct control flow graphs, helping LLMs identify bug locations more effectively. Islam et al. (Islam et al., 2024) propose MapCoder, a multi-agent framework that mimics the human coding process through retrieval, planning, coding, and debugging. It achieves state-of-the-art results on diverse programming benchmarks, showing strong generalization and robustness on complex tasks. Lin et al. (Li et al., 2025) introduces FlowGen, a multi-agent framework that simulates software process models with role-based LLMs, achieving superior code quality and stability over baselines through structured collaboration.

## B.2 PRELIMINARIES KNOWLEDGE OF GRPO

Unlike SFT, which relies on token-level loss, GRPO is a modified version of PPO that optimizes policies using policy gradients derived from reward-based losses. It encourages the exploration of richer and more diverse reasoning paths by comparing responses sampled within the same group.

Formally, let $Q$ be the question set, which contains various programming questions along with their accompanying test cases. Let $\pi_{\theta_{old}}$ be the current policy model, and $\{o_1, o_2, \ldots, o_G\}$ a group of responses generated by $\pi_{\theta_{old}}$ for question $q$. Let $\pi_{\theta_{ref}}$ denote the frozen reference model. The GRPO optimization objective is defined as follows:

$$J_{GRPO}(\theta) = \mathbb{E}_{q \sim Q, \{o_i\}_{i=1}^G \sim \pi_{\theta_{old}}}$$

$$\left[ \frac{1}{G} \sum_{i=1}^G \sum_{t=1}^{|o_i|} \min \left( \frac{\pi_\theta(o_{i,t}|q)}{\pi_{\theta_{old}}(o_{i,t}|q)} A_i, \, \text{clip} \left( \frac{\pi_\theta(o_{i,t}|q)}{\pi_{\theta_{old}}(o_{i,t}|q)}, 1 - \epsilon, 1 + \epsilon \right) A_i \right) - \beta D_{KL}(\pi_\theta \| \pi_{ref}) \right] \tag{6}$$

Here, $\epsilon$ and $\beta$ denote the clipping threshold and the coefficient for the KL-divergence penalty, respectively. The advantage $A_i$ for each response is calculated as:

$$A_i = \frac{r_i - \text{mean}(\{r_1, r_2, \ldots, r_G\})}{\text{std}(\{r_1, r_2, \ldots, r_G\})}, \quad \text{where } \{r_i\}_{i=1}^G \text{ are reward set.} \tag{7}$$

GRPO replaces the critic model used in PPO with a more efficient intra-group advantage estimation, reducing computational overhead.

## B.3 EXTRA EXPERIMENT

### B.3.1 EXPERIMENT DETAILS

In this experiment, the two agents were trained with reinforcement learning using the following hyperparameters: the learning rate (`lr`) was set to $1.0 \times 10^{-6}$, the weight decay (`weight_decay`) was $1.0 \times 10^{-2}$, the optimizer was `adamw` (choices: `adamw` or `adamw_bf16`), the learning-rate warmup ratio (`lr_warmup_ratio`) was 0, and the number of rollout samples was fixed at 5. During reinforcement learning, the planner agent was trained for 70 steps, and the coder agent for 150 steps.

For supervised fine-tuning (SFT), we used a per-device batch size of 1 with gradient accumulation over 2 steps (effective batch size 2). The learning rate was $1.0 \times 10^{-5}$ with a cosine scheduler and a 0.1 warmup ratio, and the model was trained for 1 epoch.

The reinforcement learning stage used 12,000 samples from (Xu et al., 2025; Li et al., 2023), while the SFT stage used 15,000 samples from (Xu et al., 2025; Guha et al., 2025; Li et al., 2023). Please note that neither the SFT dataset nor the RL dataset leaks into the benchmark test questions.

Table 4: The efficiency and maintainability of the generated code are evaluated using Runtime ($\downarrow$), MU ($\downarrow$), and CC ($\downarrow$), where lower values of Runtime, MU, and CC indicate better performance. **Bold** indicates the best performance for clarity.

| Method | LiveBench | | | MBPP | | | CodeContests | | | CodeForces | | |
|---|---|---|---|---|---|---|---|---|---|---|---|---|
| | Runtime | MU | CC | Runtime | MU | CC | Runtime | MU | CC | Runtime | MU | CC |
| Qwen2.5-7B-Instruct | 7.5 | 525.3 | 4.4 | 6.9 | 229.6 | 2.1 | 6.9 | 625.4 | 5.4 | **2.9** | 754.7 | **6.9** |
| AlgoForge w/ 7B-Instruct | **6.2** | **494.7** | **3.9** | 6.4 | **207.5** | **1.9** | **6.0** | 602.6 | 5.1 | 3.8 | **722.9** | 7.4 |
| Qwen2.5-7B-Coder-Instruct | 7.1 | 501.4 | 4.1 | 6.6 | 208.4 | 1.8 | 6.7 | **603.6** | 5.2 | 2.8 | 747.1 | 6.8 |
| AlgoForge w/ 7B-Coder | **6.5** | 497.5 | 3.8 | 6.4 | **202.4** | **1.7** | **6.2** | 606.7 | **5.0** | 2.8 | 727.4 | **6.6** |

Table 5: The performance of the generated code is evaluated using FR ($\downarrow$), TOE ($\downarrow$), VE ($\downarrow$), and TE ($\downarrow$), where lower values of FR, TOE, VE, and TE indicate better performance. **Bold** highlights the best performance for clarity.

| Method | LiveBench | | | | MBPP | | | | CodeContests | | | | CodeForces | | | |
|---|---|---|---|---|---|---|---|---|---|---|---|---|---|---|---|---|
| | FR | TOE | VE | TE | FR | TOE | VE | TE | FR | TOE | VE | TE | FR | TOE | VE | TE |
| Qwen2.5-7B-Instruct | 25.7 | 6.9 | 2.7 | 1.7 | 8.2 | **0.03** | 2.0 | **0.39** | 22.9 | 3.1 | 1.3 | 3.9 | 23.3 | 5.8 | 8.1 | **2.7** |
| AlgoForge w/ 7B-Instruct | **16.2** | **1.7** | **0.9** | **1.1** | 4.3 | 1.9 | **0.8** | 0.6 | **13.5** | 2.7 | **0.9** | 1.4 | **20.9** | 2.9 | 4.6 | 3.5 |
| Qwen2.5-7B-Coder-Instruct | 28.4 | **3.8** | 3.7 | 1.8 | 6.2 | **1.7** | **0.9** | 0.3 | 25.5 | 3.1 | 1.9 | **1.2** | 30.6 | 11.8 | **6.1** | 2.1 |
| AlgoForge w/ 7B-Coder | **15.9** | 5.5 | **2.7** | **1.2** | **5.9** | 2.4 | 1.2 | **0.2** | **16.7** | 2.2 | **1.5** | 2.1 | **24.7** | 10.8 | 7.2 | 1.3 |

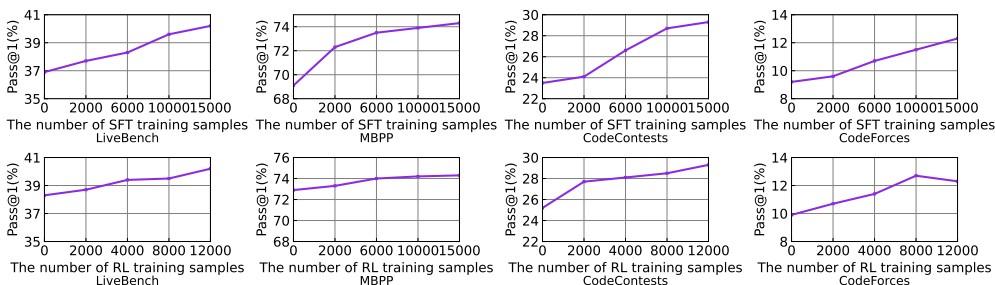

Figure 5: Impact of RL and SFT data scale on AlgoForge, where Qwen2.5-7B-Coder-Instruct is considered as base model.

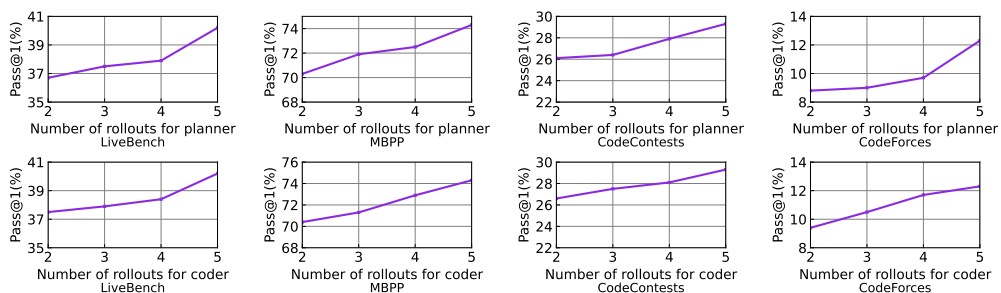

Figure 6: Impact of RL rollout numbers for planner and coder agent. Qwen2.5-7B-Coder-Instruct is considered as base model.

### B.3.2 EXTRA EXPERIMENT RESULTS

**Efficiency and maintainability of the generated code:** We evaluate the efficiency and maintainability of code generated by AlgoForge. The average runtime and average MU are used to measure time complexity and space complexity, respectively, while cyclomatic complexity (CC) is adopted as the maintainability metric. The base models are Qwen2.5-7B-Instruct and Qwen2.5-7B-Coder-Instruct, denoted as AlgoForge w/ 7B-Instruct and AlgoForge w/ 7B-Coder, and the results are reported in Table 4.

Table 4 presents the efficiency and maintainability metrics of each method on LiveBench, MBPP, CodeContests, and CodeForces. On **LiveBench**, AlgoForge w/7B-Instruct reduces runtime (7.5 s→6.2 s), MU (525.3→494.7), and CC (4.4→3.9) compared with the base model, while AlgoForge w/7B-Coder also achieves similar improvements (7.1 s→6.5 s, 501.4→497.5, 4.1→3.8).

On **MBPP**, AlgoForge w/7B-Instruct lowers runtime (6.9 s→6.4 s), MU (229.6→207.5), and CC (2.1→1.9), while AlgoForge w/7B-Coder further improves performance (6.6 s→6.4 s, 208.4→202.4, 1.8→1.7).

On **CodeContests**, AlgoForge w/7B-Instruct achieves better runtime (6.9 s→6.0 s), MU (625.4→602.6), and CC (5.4→5.1), with similar trends observed for the Coder variant.

On **CodeForces**, the improvements are smaller: runtime and CC are similar to or slightly higher than the base models. This is because AlgoForge is able to solve more complex problems, which naturally require longer execution time. In contrast, the base models fail to handle such challenging tasks and thus avoid generating code involving more sophisticated operations.

Overall, on LiveBench, MBPP, and CodeContests, AlgoForge consistently outperforms the base models, confirming that its logical enhancements lead to more efficient and maintainable code. On CodeForces, although there is a slight decline, this reflects the fact that AlgoForge is capable of solving more difficult problems rather than a loss in efficiency.

**Reliability of generated code:** To examine whether the enhanced algorithmic thought planning in AlgoForge can mitigate code generation errors, we adopt the failure rate (FR) as the evaluation metric (Wang et al., 2025b). To further refine the analysis, we focus on the proportions of the three most common error types, namely timeout errors (TOE), value errors (VE), and type errors (TE), across all generated samples. We selected Qwen2.5-7B-Instruct and Qwen2.5-7B-Coder-Instruct as the base models for AlgoForge. In Table 4, they are denoted as AlgoForge w/ 7B-Instruct and AlgoForge w/ 7B-Coder, respectively.

As shown in Table 5, both AlgoForge$_{w/7B\text{-}Instruct}$ and AlgoForge$_{w/7B\text{-}Coder}$ achieve consistently lower FR, TOE, VE, and TE than their respective base models across most benchmarks. Notably, AlgoForge$_{w/7B\text{-}Instruct}$ attains the best overall results on LiveBench and CodeContests, while AlgoForge$_{w/7B\text{-}Coder}$ also shows clear improvements on MBPP and CodeForces. These results demonstrate that our method effectively reduces errors and improves execution efficiency.

Our results demonstrate that AlgoForge framework consistently outperforms the base models across multiple benchmarks, achieving lower FR, TOE, VE, and TE. These improvements indicate that AlgoForge effectively enhances the robustness and execution capability of the generated code.

### B.3.3 SENSITIVE ANALYSIS

**Impact of data scaling on AlgoForge:** To demonstrate the impact of SFT and RL data scale on AlgoForge, we use Qwen2.5-7B-Coder-Instruct as the base model and conduct evaluations under different amounts of training data. We use Pass@1 as the evaluation metric.

Fig.5 illustrates the impact of different data scales on AlgoForge across four benchmarks. The top row shows the performance trend with increasing amounts of **SFT data**, while the bottom row shows the effect of **RL data**. As seen in the top row, increasing SFT samples consistently leads to performance gains on all benchmarks. For instance, on MBPP, Pass@1 improves from around 69 to 74, while on CodeContests and LiveBench, the gains are also substantial. This demonstrates the importance of high-quality supervised data in building a strong initial policy.

In the bottom row, RL data also contributes positively to performance, though the gains are generally smaller and tend to saturate compared to SFT. For example, on MBPP and CodeContests, Pass@1 still improves steadily with more RL samples, while on CodeForces, the curve rises and then plateaus. This suggests that while RL can refine performance, its marginal benefit diminishes with data size.

The results demonstrate that increasing both SFT and RL data scaling leads to consistent improvements in Pass@1 across all benchmarks. SFT data contributes the most significant gains by establishing a strong base performance, while RL data provides additional refinement with diminishing returns as the data scale grows. Overall, AlgoForge benefits from larger training data in both stages, achieving better code generation quality with only moderate incremental effort.

**Sensitive analysis on RL rollout:** In addition to the five RQs, the rollout sample number in the RL stage is also an important parameter. To examine its impact on accuracy, we present the relationship between the rollout sample number and Pass@1 in Table 6. It is evident that, for both the planner and the coder agents, increasing the rollout sample number consistently improves Pass@1.

### B.4 TRAINING DYNAMICS OF ALGOFORGE

### B.5 EXAMPLE OF ALGOFORGE

**AlgoForge (Sample 1)**

Q: Write a function to replace all spaces in the given string with character * list item * list item * list item * list item '

```
<plan>
Input:
- s (string): The original input string which may contain spaces.
- Output:
```

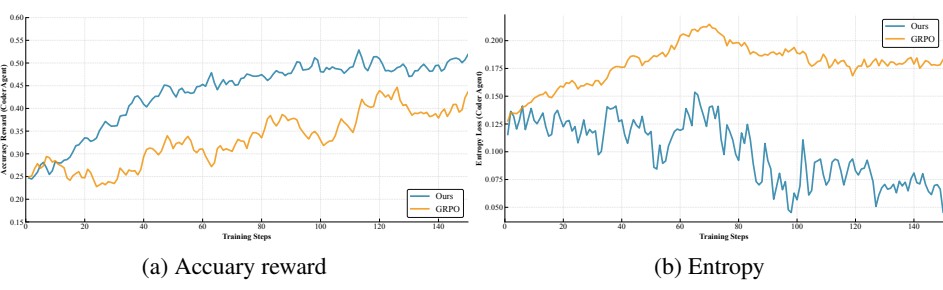

        (a) Accuary reward                        (b) Entropy

Figure 7: The training dynamics of AlgoForge, where Qwen2.5-7B-coder-Instruct is considered as basic model.

```
- A new string where every space character '' '' in s is replaced
    with the sequence ''%20''.

Step 1: Define the Input-Output Structure

 Input: A single string 's'.
 Output: A transformed string in which all space characters are
    substituted with ''%20''.

Step 2: Solving Logic

 Sequence:

 1. Initialize an empty result container (conceptually a list of
     characters or substrings).
 2. Traverse the input string from the first to the last character.

 Branch:

 1. For each character:

    If the character is a space ('' ''), append ''%20'' to the
        result.
    Else, append the character itself unchanged.

 Loop:

 1. A loop is required to process each character in the input
     string sequentially until the end is reached.

 Final Step:

 1. Combine all elements in the result container into a single
     string.
 2. Return this new string.
</plan>
```

```python
def replaceSpaces(s):
    result = []
    for char in s:
        if char == ' ':
            result.append("%20")
        else:
            result.append(char)
    return "".join(result)
```

**AlgoForge** (Sample 2)

Q: Write a function to get a lucid number smaller than or equal to n.

```
<plan>
Input:
- n (an integer): The upper bound for searching lucid numbers, where
    n >= 1.

Output:

The largest lucid number less than or equal to n.

1. Define Lucid Number:

   A lucid number is a positive integer whose prime factors are only
       2, 3, 5, or 7.
   Examples: 1, 2, 3, 5, 7, 6, 10, 14, 15, 21, 28, etc.

2. Sequence (Step-by-Step Process):

   1. Start with the base set {1}, since 1 is trivially lucid.
   2. Generate new lucid numbers by multiplying existing ones by 2,
       3, 5, or 7.
   3. Continue generating until values exceed n.
   4. Store all generated lucid numbers in a data structure (e.g.,
       list or heap).
   5. Select the maximum lucid number that is <= n.

3. Branch (Conditions):

   If n < 1 -> no lucid number exists, return 0 or handle as invalid
       input.
   If n itself is lucid -> directly return n.
   Otherwise, search among generated lucid numbers for the largest
       <= n.

4. Loop (Repetition):

   Use a loop or priority-based expansion to multiply numbers
       repeatedly until exceeding n.
   Continue until no new lucid numbers can be generated under the
       bound.
</plan>
```

```python
def largest_lucid_number(n):
    if n < 1:
        return 0
    primes = [2, 3, 5, 7]
    lucid = {1}
    queue = [1]

    while queue:
        current = queue.pop()
        for p in primes:
            new_val = current * p
            if new_val <= n and new_val not in lucid:
                lucid.add(new_val)
                queue.append(new_val)

    return max(x for x in lucid if x <= n)
```

**AlgoForge** (Sample 3)

Q: Due to the recent popularity of the Deep learning new countries are starting to look like Neural Networks. That is, the countries are being built deep with many layers, each layer possibly having many cities. They also have one entry, and one exit point.
There are exactly L layers, each having N cities. Let us look at the two adjacent layers L1 and L2. Each city from the layer L1 is connected to each city from the layer L2 with the traveling cost cij for
$$i, j \in \{1, 2, \ldots, N\}$$
, and each pair of adjacent layers has the same cost in between their cities as any other pair (they just stacked the same layers, as usual). Also, the traveling costs to each city from the layer L2 are same for all cities in the L1, that is cij is the same for
$$i \in \{1, 2, \ldots, N\}$$
, and fixed j.
Doctor G. needs to speed up his computations for this country so he asks you to find the number of paths he can take from entry to exit point such that his traveling cost is divisible by given number M.

```
<plan>
- L (integer): number of layers in the country (excluding entry/exit
    ).
- N (integer): number of cities in each layer.
- c (list of integers of length N): traveling costs to each city in
    the next layer. For all i in layer L1, cost to city j in layer
    L2 is `c[j]`.
- M (integer): the modulus divisor.

Output:

 count (integer): the number of distinct paths from entry to exit
    such that the total traveling cost is divisible by M.

Strategy:

1. Input-Output Structure

   Input defines a stacked structure of L layers, each with N cities
       , connected in a complete bipartite way to the next layer
       with uniform costs per destination.
   Output is a single integer, representing the number of valid
       paths modulo or exact (depending on constraints).

2. Solving Logic

   Sequence:

    1. Model path costs as sequences of choices across L transitions
         (from entry to exit).
    2. Since costs depend only on the chosen city in each layer, the
         total cost is the sum of L chosen city costs (one from each
         layer).
    3. The number of paths equals the number of such sequences,
         which is $N^L$. But we only count sequences whose total cost
         % M == 0.
   Branch:

    For each possible remainder modulo M, decide whether adding a
        new city's cost keeps us on a path toward divisibility.
    Use a dynamic programming (DP) state:
```

```
       `dp[layer][r] = number of ways to reach remainder r after
           processing that many layers`.
    Loop:

     For each layer from 1 to L:

       For each remainder r:

         For each city j in that layer (N choices):

           Update next remainder: `(r + c[j]) % M`.
           Accumulate counts accordingly.
</plan>
```

```python
def count_paths(L, N, costs, M):
    # Precompute frequencies of costs modulo M
    freq = [0] * M
    for c in costs:
        freq[c % M] += 1

    # Initialize DP: only 0 remainder has one path (empty path)
    dp = [0] * M
    dp[0] = 1

    # Process each layer
    for _ in range(L):
        new_dp = [0] * M
        for r in range(M):
            if dp[r] == 0:
                continue
            for mod_val in range(M):
                if freq[mod_val] == 0:
                    continue
                new_r = (r + mod_val) % M
                new_dp[new_r] += dp[r] * freq[mod_val]
        dp = new_dp

    # Answer is number of ways with remainder 0
    return dp[0]
```

