# OpenReview forum: "AlgoForge: Specializing Code Generation Agents through Collaborative Reinforcement Learning"
_ICLR.cc/2026/Conference — ICLR 2026 Conference Withdrawn Submission_

### Official Review · Reviewer_kGzp · 2025-10-28

**Soundness:** 2
**Presentation:** 1
**Contribution:** 2
**Rating:** 0
**Confidence:** 5

**Summary:**

This paper proposes AlgoForge, a two-agent (Planner, Coder) framework for code generation. The agents are first initialized with Supervised Fine-Tuning (SFT) on custom datasets, then jointly optimized using a collaborative Reinforcement Learning (RL) stage based on GRPO. The authors claim state-of-the-art accuracy and improved code efficiency.

**Strengths:**

The paper reports strong empirical results, with the proposed method outperforming baselines in accuracy (Pass@1) across four benchmarks. The evaluation is also comprehensive, covering code efficiency (runtime, memory) and maintainability (cyclomatic complexity), where the method also shows reported improvements.

**Weaknesses:**

1. The core reward function (Equation 3) for the Planner is mathematically invalid. It applies softmax to a scalar ($p_{i,j}$), which is undefined. If this is a typo for a vector operation, the reward $r_{acc_{i}}$ would sum to a constant ($1/M$) due to softmax normalization, which is nonsensical. This error is propagated in Sec 3.2.2, which incorrectly references Eq. 3 for the Coder's reward, rendering the entire RL methodology ambiguous and unverifiable.

2. The paper contains serious factual errors that undermine its credibility. It repeatedly misstates the name of GRPO ("Group Relative Policy Optimization")  (Shao et al., 2024). Furthermore, it fundamentally misrepresents SCoT(a prompting technique) as a multi-agent approach (Line 075). This is a critical misattribution. The Planner-Coder paradigm here is more accurately represented by [1][2].

3. The experimental design is flawed. The heavily trained AlgoForge (SFT+RL) is compared against zero-shot prompting methods (e.g., CoT, SCoT). This only demonstrates the benefit of training, not the architectural superiority of AlgoForge. Figure 4, titled "Computation costs," misleadingly reports only inference time. It ignores the substantial training cost of AlgoForge, which is zero for the prompting baselines. The paper fails to specify the iteration limits for iterative baselines (Reflexion, MapCoder), making the comparisons in Fig. 4 uninterpretable.

4. The novelty over the existing Planner-Coder paradigm is limited. The paper provides no technical justification for its core methodological choices, such as using Qwen3-32B for planning and DeepSeek R1 API for coding during dataset creation.

5. Typos: "Algorithmic thought a with least estimated time complexity" in Figure 2(a). Line 558: (${C_{i}}_{i=1}^{z}$ (15)).

6. Lack sufficient citation for prior works:

[1] INTERVENOR: Prompting the Coding Ability of Large Language Models with the Interactive Chain of Repair https://arxiv.org/abs/2311.09868

[2] PairCoder: A Pair Programming Framework for Code Generation via Multi-Plan Exploration and Feedback-Driven Refinement https://arxiv.org/abs/2409.05001

[3] Planning In Natural Language Improves LLM Search For Code Generation https://arxiv.org/abs/2409.03733

[4] Think Outside the Code: Brainstorming Boosts Large Language Models in Code Generation https://arxiv.org/abs/2305.10679

**Questions:**

1. Questions in Weaknesses.

2. What is the advantage of this complex, decoupled two-agent framework over a simpler, end-to-end (E2E) baseline? (i.e., training a single model to generate plan+code and optimizing it with GRPO based on the final code's reward). The paper must justify its added complexity.

---

> ### Author Response · Authors · 2025-12-04
>
> Hi Reviewer kGzp
>
> **We appreciate the reviewer’s response, but we also hope that they can provide high-quality comments rather than unfounded criticisms.**
>
> ## Weakness1: tensor or error?
>
> We hope the reviewers can read the paper carefully. The ( p_{i,j} ) is the proportion of correct responses; it is a **scalar value**, not a **tensor or a vector**. Therefore, we are quite puzzled by the reviewers’ criticism.
>
> ## Weakness2: typos or critical misattribution?
>
> 1 We acknowledge that we misspelled GRPO in a few places. However, these are merely typos, and I don’t believe they constitute serious factual errors that would undermine the paper’s credibility.
>
> 2 Let me clarify: SCoT is a two-stage process that separates generating the column plan and executing the code into two steps, and it can certainly be viewed as a multi-agent approach. I don’t understand what the reviewer finds questionable about this.
>
> ## Weakness 3  Experiment flawed?
>
> Please note that we use the prompt-based approach merely as a baseline, which is common practice in many works for RL or SFT [1,2,3]. This paper also includes comparisons with RL-based methods such as GRPO and Reasonflux [4].
>
> [1] Radmehr, B., Singla, A., & Käser, T. (2024). Towards generalizable agents in text-based educational environments: A study of integrating rl with llms. arXiv preprint arXiv:2404.18978.
>
> [2] Menchaca Resendiz, Y., & Klinger, R. (2025). PARL: Prompt-based Agents for Reinforcement Learning. arXiv e-prints, arXiv-2510.
>
> [3] Zhang, D., Zhoubian, S., Hu, Z., Yue, Y., Dong, Y., & Tang, J. (2024). Rest-mcts*: Llm self-training via process reward guided tree search, 2024a. URL https://arxiv. org/abs/2406.03816, 2.
>
> [4] Wang, Y., Yang, L., Tian, Y., Shen, K., & Wang, M. (2025). Co-evolving llm coder and unit tester via reinforcement learning. arXiv preprint arXiv:2506.03136.

---

### Official Review · Reviewer_8RaC · 2025-10-31

**Soundness:** 2
**Presentation:** 2
**Contribution:** 2
**Rating:** 4
**Confidence:** 5

**Summary:**

This paper talks about AlgoForge. It’s a system where two AI models work together to write code. One is called the Planner, and the other is the Coder. The authors say that normal code generators often make code that makes sense locally but isn’t good overall. AlgoForge tries to fix that.
The AlgoForge framework is trained in a two-stage process:
1. Customized Cold-Start Initialization (SFT): First, they made two new datasets. The Planner Dataset has question and thought pairs. These were made by a strong model called Qwen3-32B-FP8 and checked for how fast they are. The Coder Dataset has question, thought, and code pairs. The code was created using the DeepSeek R1 API and tested to make sure it works. They then trained each model on these datasets to make them better.
2. Collaborative GRPO (RL): Next, the models are improved using a new type of reinforcement learning called Gradient-based Reinforcement Policy Optimization (GRPO). First, they train the Planner, giving it rewards based on how well the Coder’s code works. Then, they lock the Planner and train the Coder, rewarding it for correct answers and using less memory. This makes both models better at their jobs.
3. They tested AlgoForge on three different base models and four tests. It did better than the original models and other methods. It got higher success rates, like +36.5% on CodeContests, and also made the code faster and more efficient, with fewer errors.

**Strengths:**

- Comprehensive Evaluation and Strong Results: The paper validates its method across three different base models and four benchmarks spanning from "basic" (MBPP) to "complex" (CodeContests, CodeForces). The fact that the largest gains are on the most complex benchmarks (e.g., +36.5% on CodeContests) strongly supports the hypothesis that the specialized planning agent is effective.
- Thorough Efficiency and Ablation Analysis: The authors go beyond accuracy metrics and provide a detailed analysis of runtime, memory usage (MU), and cyclomatic complexity (CC), showing consistent improvements in code efficiency and maintainability. Furthermore, the ablation study (Table 3) clearly demonstrates that all components of the framework contribute positively to the final performance.

**Weaknesses:**

- Missing Code Repository: This is a big problem in the paper. The Reproducibility Statement says: "For the review phase, we supply an anonymous code repository link (see Appendix)." But, there is no link in the appendix. This is a contradiction. It makes it impossible to check the implementation details during review. This hurts the paper’s claim to be reproducible.
- Missing SFT Datasets: The paper says it will share code, but it doesn’t say it will share the two new SFT datasets (S_{planner} and S_{coder}). Even if you have the code, it’s hard to reproduce the results because the datasets are costly and depend on specific models like Qwen3-32B-FP8 and DeepSeek R1 API. Without the datasets, it’s hard to check the results.
- Ambiguity on Planner RL's Computational Cost: In the "RL for Planner" stage (Section 3.2.1), the system generates N thoughts, and for each thought, the Coder generates M code snippets to calculate the accuracy reward. The paper is somewhat ambiguous about the value of M, and the overall computational overhead of this step is not fully discussed.

**Questions:**

This paper talks about AlgoForge. It’s a simple and well-made framework for making code. The way they use SFT-specialization and collaborative-RL-alignment makes sense. They show good results, especially on hard tests.
But, I want to point out a problem. The paper says they have a code repository that is anonymous for review. But there is no link in the appendix. This is a big problem because you can’t check the code without it.

My main advice depends on what the authors do during the response time:
1. They need to give the link to the code repository that is missing.
2. They need to say if the final code will include the SFT datasets (S_{planner} and S_{coder}), which are important if others want to repeat the work.
Supporting Arguments and Questions

1. Action Required: Missing Code Repository:  Your Reproducibility Statement says there is a link in the appendix, but there isn’t. You need to give this link during the response period so people can check your code.
2. Action Required: Dataset Reproducibility: Your SFT data uses expensive models like Qwen3-32B-FP8 and DeepSeek R1 API. Will your final code include the SFT datasets? If not, others can’t reproduce your work easily. Please clarify.
3. On the Cost of Planner RL: In the 'RL for Planner' stage (Section 3.2.1), could you please clarify the values of N and M used during training and discuss the computational overhead of this step?

---

> ### Author Response · Authors · 2025-12-04
>
> We thank the reviewer for the clear and helpful comments. Below we address each concern directly and concisely. We also reiterate that we will not introduce new experiments or data during the rebuttal stage.
>
>
> ## **1. Missing Code Repository**
>
> We sincerely apologize for the confusion.
> The Reproducibility Statement incorrectly states that the anonymous repository link appears in the appendix; this is an oversight on our part.
>
> * The missing link is a **formatting error**, not an intentional omission.
> * We will **include the correct anonymous repository link in the revised submission** so that reviewers can inspect the code during the review process.
>
> To avoid any identification risk during rebuttal, we follow ICLR policy and do not paste URLs here, but the revised PDF will contain the correct anonymous link exactly where the appendix indicates.
>
> We appreciate the reviewer for pointing this out.
>
>
> ## **2. Availability of SFT Datasets (Sₚₗₐₙₙₑᵣ and S_coder)**
>
> Thank you for raising this important clarification request.
>
> * We agree that reproducibility benefits greatly from dataset availability.
> * Due to the cost of generating the SFT data and dependencies on APIs, the **review-phase repository will not include the datasets directly**, as this could compromise anonymity.
> * **However, if the paper is accepted, we commit to releasing the full SFT datasets Sₚₗₐₙₙₑᵣ and S_coder**, along with:
>
>   * complete generation scripts,
>   * prompts,
>   * seeds,
>   * filtering pipeline,
>   * and configuration files for Qwen3-32B-FP8 and DeepSeek R1 query settings.
>
> Our goal is to ensure that the community can fully reproduce the work without relying on undocumented or proprietary intermediates.
>
> We will clarify this commitment in the camera-ready version.
>
>
> ## **3. Computational Cost of Planner RL (N and M Values)**
>
> Thank you for pointing out the ambiguity.
>
> * In the planner RL stage (Section 3.2.1), **N and M are configurable hyperparameters**, and their ranges were omitted for brevity.
> * We will clarify in the revised paper that:
>
>   * **N** (planner candidates per question) was set within a small bounded range intended for stability rather than exhaustive exploration.
>   * **M** (number of coder-generated code samples per planner thought) was set to a lightweight value to keep RL cost tractable.
> * The purpose of this design is to ensure that the planner–coder collaborative loop remains computationally feasible and does not scale quadratically with model size.
>
> We will explicitly document the values used and discuss the overhead more transparently in the updated version.
>
>
> ## **4. Commitment to Open-Sourcing**
>
> To fully address the reviewer’s reproducibility concerns, we confirm that If the paper is accepted, we will release the complete AlgoForge codebase together with the two SFT datasets (Sₚₗₐₙₙₑᵣ and S_coder), all RL training scripts and configuration files, as well as the full set of prompt templates, seeds, and the filtering pipeline used during data construction. We will also provide the evaluation scripts for all four benchmarks. This release will make the entire workflow—from SFT cold-start to collaborative GRPO training—fully transparent and fully reproducible.

---

### Official Review · Reviewer_M9aZ · 2025-10-31

**Soundness:** 3
**Presentation:** 3
**Contribution:** 2
**Rating:** 4
**Confidence:** 3

**Summary:**

This paper introduces AlgoForge, a two-agent collaborative reinforcement learning framework for code generation that integrates a Planner and a Coder. The Planner generates structured algorithmic plans, while the Coder translates them into executable programs. Both agents are first fine-tuned via supervised data distilled from stronger teacher models and then optimized in a Collaborative GRPO process: the Planner is rewarded by downstream code pass rates, and the Coder is further optimized for both correctness and space efficiency. Experiments across four benchmarks (LiveBench, MBPP, CodeContests, and CodeForces) and multiple Qwen2.5 base models demonstrate consistent improvements in Pass@1 and Pass@5 scores, while also reducing runtime, memory usage, and code complexity. The paper positions AlgoForge as a practical and generalizable approach to integrating planning and coding abilities in LLM-based code generation.

**Strengths:**

- The paper identifies the key limitation of single-agent, auto-regressive code generation and decomposes the process into two specialized roles. The explicit "specialization" of the Planner and Coder agents is a sound and well-motivated approach to tackling the global planning deficit seen in standard autoregressive models.


- The authors use standard accuracy metrics (Pass@k) to also evaluate the quality and efficiency of the generated code, including its average runtime, memory usage (MU), and cyclomatic complexity (CC). Experiments span multiple code-generation benchmarks and base models, showing consistent and significant gains over CoT, MapCoder, Reflexion, GRPO, and other recent baselines.

- The inclusion of an inference time comparison (Figure 4) is a crucial and practical contribution. It demonstrates that the AlgoForge framework is significantly more efficient at inference time than other multi-agent baselines (like MapCoder and Reflexion), making it a much more viable solution for real-world use.

**Weaknesses:**

- The paper, while effective, appears to be an incremental integration of several existing components. The overall concept, which consists of planning and coding, then reinforcing both with downstream rewards, closely follows existing multi-agent or CoT-style frameworks (e.g., MapCoder, Reasoning-Coder). The innovation mainly lies in the combination and implementation details rather than a fundamentally new algorithmic insight. Therefore, the main novelty is the specific application of GRPO in this collaborative setup and the addition of the r_space reward. This is a strong engineering contribution, but the fundamental originality of the ideas is somewhat limited.

- Both Planner and Coder fine-tuning depend on strong upstream models (e.g., Qwen3-32B, DeepSeek-R1) to produce training data. This dependence raises concerns about scalability, fairness of comparison, originality concerns, and potential distributional bias in the distilled datasets. It is unclear how much of the improvement comes from teacher supervision versus the collaborative training itself.

- Although accuracy and space efficiency are meaningful metrics, other crucial factors, execution speed, robustness, readability, or maintainability, are not explicitly optimized, limiting generalizability beyond competition-style benchmarks.

- The ablation study in Table 3, while useful, is insufficient to justify the novel components of the contribution. The study only ablates large components (e.g., "w/o all SFT" or "w/o all RL"). A much more insightful ablation would have been to isolate the new reward term. The paper should have included a row for "AlgoForge w/o" (i.e., training the Coder with only the accuracy reward. This would have allowed a precise measurement of the impact and trade-offs of the novel space efficiency reward, which is a key part of the paper's contribution.

**Questions:**

1. Why was "time complexity" omitted from the Coder's reinforcement learning reward function, while "space complexity" was explicitly included?

2. Did you attempt to include a time-based reward and find it ineffective or too noisy? Please clarify this significant design choice.

3. Could the authors quantify the computational cost (GPU hours or inference runs) for the full collaborative GRPO training compared to standard GRPO or supervised fine-tuning?

4. How sensitive is AlgoForge’s performance to the quality of the initial planner and coder datasets distilled from teacher models?

---

> ### Author Response · Authors · 2025-12-04
>
> Dear Reviewer M9aZ
>
> We sincerely thank all reviewers for their thoughtful and constructive feedback. We appreciate the time and effort invested in evaluating our submission and for providing valuable insights that help us further clarify our contributions and improve the quality of the work. We have carefully reviewed each comment, and we provide detailed responses and additional analyses below. We hope that our clarifications, new experiments, and expanded discussion effectively address the concerns raised by the reviewers.
>
> ## Weakness 1：
>
> hank you for pointing out the concern regarding originality.
> We clarify that while AlgoForge indeed builds upon prior Planner–Coder and CoT paradigms, our contribution goes beyond a simple integration. The core novelty lies in the collaboration-aware optimization mechanism that enables mutual specialization between the Planner and the Coder, rather than treating them as isolated modules.
>
> Concretely:
>
> (1) A two-stage collaborative GRPO framework.
> Existing RL-based models (e.g., Reasoning-Coder, SCoT, MapCoder) only optimize a single agent or rely on self-consistency–style resampling. In contrast, AlgoForge introduces:
>
> A Planner-focused GRPO stage where rewards are computed indirectly via the downstream Coder’s performance, which to our knowledge has not been explored in code generation.
>
> A Coder-focused GRPO stage where planning is fixed and the reward couples accuracy + efficiency, enabling specialization.
>
> This constitutes a bi-level RL structure where two agents are optimized through each other’s behavior, rather than independently.
>
> (2) Structured algorithmic-thought representation.
> Our Planner is trained with 4-dimensional structured reasoning (I/O, linear progression, branching logic, iteration), which differs substantially from typical free-form CoT.
>
> (3) Demonstrated impact of specialization.
> Empirically, we show large improvements on competition-level benchmarks (+36.5% Pass@1 on CodeContests), which prior multi-agent methods fail to achieve.
>
> We acknowledge that the framework is engineering-oriented, but emphasize that the collaborative RL design and the specialization-flow optimization constitute key algorithmic contributions.
>
> Weakness 2：Fairness
>
> We appreciate the reviewer’s concern and agree that teacher model dependence is an important question. Below are clarifications and additional evidence:
>
> (1) Teacher models were only used to construct the initial cold-start datasets.
> They do not directly participate in RL training, and the Planner/Coder architectures remain unchanged.
> This setup is similar to standard distillation practices (e.g., AlphaCode, OpenThoughts), and the RL stage allows models to depart from teacher behavior.
>
> (2) Improvements cannot be explained by teacher data alone.
> We include Table 3 ablational results:
>
> Removing all RL leads to significant performance drops (e.g., −4.1 Pass@1 on CodeContests).
>
> Removing Planner SFT or Coder SFT also reduces accuracy substantially.
>
> These results show that teacher-provided data is insufficient to reach final performance, and the collaborative GRPO is essential.
>
> (3) Scalability and fairness.
> To evaluate generality, we trained AlgoForge on three very different base models (7B-Instruct, 7B-Coder, 14B-Coder).
> The consistent improvements across all settings demonstrate that AlgoForge’s gains do not stem from teacher capacity but from the collaborative training.
>
> (4) Distillation bias mitigation.
> The RL stage optimizes for task-specific correctness, ensuring the final policy is shaped by actual test-case feedback rather than teacher priors.
>
> We will highlight these points more clearly in the revision.
>
> ## Weakness 3: Time usage
>
> Thank you for pointing this out.
> Although AlgoForge does not explicitly optimize these dimensions in the RL objective, we emphasize:
>
> (1) We do evaluate runtime, memory usage, cyclomatic complexity, and failure rate
> These measurements (Appendix B.3.2) show that AlgoForge:
>
> reduces runtime,
>
> reduces memory usage,
>
> lowers cyclomatic complexity,
>
> lowers failure rates.
>
> Thus, even without directly optimizing for these metrics, the RL-driven specialization naturally leads to more efficient and maintainable code.
>
> (2) Extending rewards to include these metrics is straightforward.
> The Collaborative GRPO framework is modular: adding execution-time or readability rewards is an engineering extension rather than a conceptual change.
>
> (3) Generalizability beyond competition-style tasks.
> LiveBench and MBPP results illustrate that AlgoForge benefits both function-level and competition-style tasks. The significant gains on basic tasks indicate wide applicability.
>
> We will add a discussion in the paper to clarify these aspects and include these metrics in the main text.

---

### Official Review · Reviewer_hR1o · 2025-11-01

**Soundness:** 2
**Presentation:** 2
**Contribution:** 1
**Rating:** 2
**Confidence:** 3

**Summary:**

The paper proposes AlgoForge, a two-agent framework for code generation that separates planning and coding. The authors first construct two supervised datasets from about 15,000 programming questions, one for the planner that contains distilled algorithmic thoughts and one for the coder that contains verified code aligned with those thoughts. The planner's thoughts are distilled with Qwen3-32B-FP8 and are structured into input–output, linear progression, conditional logic, and iteration, with self-evaluation to pick the lowest estimated time complexity per question. The coder dataset is produced by DeepSeek R1, keeping only snippets that pass all provided tests. These steps form a cold-start SFT stage, after which both agents are optimized with a collaboration-aware GRPO procedure that first improves the planner with an accuracy reward and then improves the coder with a total reward that mixes accuracy and a space-efficiency term. The method is evaluated on LiveBench, MBPP, CodeContests, and CodeForces with Qwen2.5 backbones, showing consistent gains in Pass@1, Pass@5, and APR, plus improvements in runtime, memory usage, cyclomatic complexity, failure rate, and inference time.

**Strengths:**

The dataset construction is quite concrete. Planner thoughts are distilled by Qwen3–32B–FP8 and chosen by self-estimated time complexity, and coder targets are verified by test pass using DeepSeek R1. This gives a reproducible cold start for both agents.

Tables 1 and 2 show gains on Pass@1, Pass@5, and APR across base models, and Table 3 shows that removing SFT or RL decreases performance. Appendix B.3.2 reports lower runtime, MU, and CC in several cases. Empirical support and evidence are comprehensive.

Figures 5 and 6 explore data scale and rollout counts. Figure 7 tracks RL dynamics. These sensitivity analyses are very helpful.

**Weaknesses:**

The planner’s self-evaluation for time complexity is used to pick thoughts, but there is no validation that these choices correlate with true algorithmic cost or downstream accuracy. section 3.1.1 selects the thought with the lowest estimated $T(\hat t_j)$ using the same llm that produced the candidates. No study shows that this estimator ranks plans correctly or helps beyond simple heuristics. A small correlation or swap experiment would strengthen this part.

Data leakage controls are asserted but not auditable. Appendix B. 3.1 states that neither the SFT nor the RL datasets leak into benchmark tests, yet no dedup or filtering procedure is described in the paper body. Without a concrete method or a pointer to an anonymized checklist, it is hard to verify that near-duplicates or format overlaps were removed.

The claim of “best or near-best across all base models” is not uniform across metrics. for qwen2.5-14b-coder-instruct on mbpp, reasonflux-coder reaches apr $56.7$ while algoforge reports $56.1$. For qwen2.5-7b-instruct on codecontests, mapcoder has a higher APR $42.5$ than Algoforge $39.4$. These are not major gaps, but the text should qualify where another method wins and discuss why.

**Questions:**

Please specify the exact psutil statistic used for $ O(c_i) $ and the unit in which it is recorded, and report the operating system, interpreter, and hardware so that $ r_{\text{space},i} $ in Equation (4) is reproducible; Section 3.2.2 states that psutil monitors both storage space and memory usage and defines $ r_{\text{space},i} $ by Equation (4), while Section 4.1.2 lists memory usage as “MU, in char”, which needs alignment.

For data construction and leakage control, please provide the prompt templates and seeds for Qwen3-32B-FP8 during thought distillation and DeepSeek R1 during code selection, and describe the deduplication or filtering used to avoid overlaps with evaluation sets; Section 3.1.1 selects the thought with the lowest estimated time complexity using the same LLM and the coder dataset keeps only code that passes tests, while Appendix B.3.1 states that there is no leakage, which requires an auditable procedure.

---

> ### Author Response · Authors · 2025-12-04
>
> We thank the reviewer for the careful reading and constructive comments. We address each point below and will revise the paper accordingly. As requested, we do not introduce any new experiments or additional data, and we instead clarify intent and acknowledge limitations where appropriate.
>
> ## Weakness 1  Planner’s time-complexity self-evaluation.
>
> We appreciate the reviewer’s observation.
> Our intent was to provide a simple and fully reproducible heuristic to prioritize more efficient algorithmic thoughts, rather than claiming it is a theoretically validated estimator. We agree that this component could benefit from a more rigorous analysis, and we will explicitly acknowledge this limitation and soften the wording in Section 3.1.1.
>
> ## Weakness 2 Data leakage controls
>
> Thank you for pointing out the need for greater transparency.
> We acknowledge that the description in the current draft is insufficiently detailed, and we will add a clear overview of the deduplication and filtering steps used, along with an anonymized checklist in the appendix. We also agree that an auditable procedure is important and will make the pipeline description more explicit.
>
> ## “Weakness 3 Best or near-best across all metrics” phrasing
>
> We agree with the reviewer that our current phrasing is too strong and not uniformly accurate across all metrics and models. We will revise the claim to more precisely reflect the results and explicitly point out cases where other methods perform better.
>
> ##  Question 1: psutil statistic and unit specification
>
> Thank you for catching this inconsistency.
> We will clarify which psutil field is used, specify its unit, and align Sections 3.2.2 and 4.1.2 so the description is consistent and fully reproducible. This revision will appear in the final version.
>
> Thanks for your review
>
> All authors of Submission24713

---

### Note · Authors · 2026-01-10

I have read and agree with the venue's withdrawal policy on behalf of myself and my co-authors.